# The CASPAR study protocol. Can cervical stiffness predict successful vaginal delivery after induction of labour? a feasibility, cohort study

**Elizabeth Medford**[1,2] *, **Steven Lane**[3], **Abi Merriel**[2,3], **Andrew Sharp**[1,2], **Angharad Care**[1,2]

1 Department of Women and Children's Health, Harris Preterm Birth Research Centre, University of Liverpool, Liverpool, United Kingdom, 2 Liverpool Women's Hospital, NHS Foundation Trust, Liverpool, United Kingdom, 3 University of Liverpool, Liverpool, United Kingdom

* emedford@liverpool.ac.uk

**Data Availability Statement:** No datasets were generated or analysed during the current study. All relevant data from this study will be made available upon study completion.

## Abstract

### Background

Induction of labour (IOL) is a common obstetric intervention in the UK, affecting up to 33% of deliveries. IOL aims to achieve a vaginal delivery prior to spontaneous onset of labour to prevent harm from ongoing pregnancy complications and is known to prevent stillbirths and reduce neonatal intensive care unit admissions. However, IOL doesn't come without risk and overall, 20% of mothers having an induction will still require a caesarean section birth and in primiparous mothers this rate is even higher.

There is no reliable predictive bedside tool available in clinical practice to predict which patient's undergoing the IOL process will result in a vaginal birth; the fundamental aim of the IOL process. The Bishop's Score (BS) remains in routine clinical practice as the examination tool to assess the cervix prior to IOL, despite it being proven to be ineffective as a predictive tool and largely subjective. This study will assess the use of the Pregnolia System, a new objective antenatal test of cervical stiffness. This study will explore its' potential for pre-induction cervical assessment and indication of delivery outcome following IOL.

### Methods

CASPAR is a feasibility study of term, primiparous women with singleton pregnancies undergoing IOL. Cervical stiffness will be assessed using the Pregnolia System; a novel, non-invasive, licensed, CE-marked, aspiration-based device proven to provide objective, quantitative cervical stiffness measurements represented as the Cervical Stiffness Index (CSI, in mbar). A measurement is obtained by applying the sterile single-use Pregnolia Probe directly to the anterior lip of the cervix, visualised via placement of a speculum.

Following informed consent, CASPAR study participants will undergo the Pregnolia System cervical stiffness assessment prior to their IOL process commencing. Participant questionnaires will evaluate the acceptability of this assessment tool in this population. This

**Funding:** Pregnolia AG provided the device for use in this study and provided financial aid to fund a Clinical Research Fellow to undertake this research. Pregnolia AG had no role in the study design, decision to publish or preparation of the manuscript.

**Competing interests:** The authors have declared that no competing interests exist.

study will directly compare this novel antenatal test to the current BS for both patient experience of the different cervical assessment tools and for IOL outcome prediction.

## Discussion

This feasibility study will explore the use of this novel device in clinical practice for pre-induction cervical assessment and delivery outcome prediction. Our findings will provide novel data that could be instrumental in transforming clinical practice surrounding IOL. Determining recruitment rates and acceptability of this new assessment tool in this population will inform design of a further powered study using the Pregnolia System as the point-of-care, bedside cervical assessment tool within an IOL prediction model.

## Study registration

This study is sponsored by The University of Liverpool and registered at ClinicalTrials.gov, identifier NCT05981469, date of registration 7th July 2023.

## Introduction

Induction of labour (IOL) is common in obstetrics affecting up to 33% of deliveries in the UK [1]. It's indicated where the expected benefit of expediting labour outweighs the potential harms of awaiting spontaneous onset and affects the birth options and birthing experience of the woman [2–4]. IOL is undertaken with the intention to achieve a vaginal birth, yet over 20% of inductions will ultimately require an unplanned caesarean section which has additional maternal and neonatal risks, as well as an increase in cost and use of healthcare resources [1,5,6]. There is no current recommended predictive model to determine which women undergoing IOL will achieve a vaginal birth [7,8].

At an individual level, a verified and reliable prediction model would be instrumental in providing women and their clinicians with robust information to allow more informed, decision making surrounding the risks associated with IOL care; importantly, their individualised risk of unsuccessful IOL and requirement for delivery by caesarean section[9,10]. At a policy level, a successful IOL prediction model could improve maternity service costing and resource allocation planning in an already overstretched and underfunded healthcare service [11,12].

The IOL process requires changes in cervical status from a quiescent pregnant state to an actively labouring state. Previously explored clinical assessment tools to assess cervical change ready for labour include digital palpation; Bishop's score (BS) and transvaginal ultrasound (USS); cervical length, cervical angle, cervical funnelling and cervical elastography[13–19]. In current literature these assessment tools have not been recommended for routine clinical use in IOL prediction models either due to subjectivity, poor predictive value or expensive and labour-intensive resources required[18–24]. Thus the clinical question remains as to how best assess the cervix prior to IOL to subsequently predict the outcome.

The Pregnolia System is a CE- certified, novel device that provides objective, quantitative assessment of cervical stiffness through an aspiration-based technique[25,26]. A measurement is obtained by applying the device directly to the anterior lip of the cervix, visualised via placement of a speculum, and gives a quantitative assessment of cervical stiffness represented as the Cervical Stiffness Index (CSI, in mbar). It has been proven to be superior to digital palpation in an in-vitro setting[27] and has already shown promise in a clinical setting as a tool for

preterm birth prediction[28–30]. A feasibility study is needed to explore the application of this novel device to the IOL setting for pre-induction cervical assessment. Assessing the acceptability, implementation and efficacy of the Pregnolia System for consistent cervical stiffness assessments in this population is an exciting prospect to inform further IOL prediction model trial design.

This feasibility study will explore whether cervical stiffness obtained using the Pregnolia System has an association with the outcome of IOL in term, primiparous women. It will directly compare current routine clinical practice of BS to the cervical stiffness obtained using the Pregnolia System, and explore the association with vaginal delivery as an outcome of IOL [2]. This study will assess the ability to use this device in real clinical practice and explore the acceptability of the Pregnolia System as an assessment tool in this population for this purpose. Together, these results will help inform future study design using the Pregnolia System as the simple, point-of-care, bedside cervical assessment tool within an IOL prediction model.

## Methods and materials

### Study design

This feasibility study is a single site prospective, cohort study of primiparous women with a singleton pregnancy undergoing IOL at the Liverpool Women's Hospital in the UK. Participants will be approached to participate at the time of attendance for their planned IOL as decided by their clinical team.

As per NICE guidelines they will have routine procedures prior to IOL including; confirmation of cephalic presentation and a CTG to confirm normal fetal heart rate and absence of uterine activity [31]. Following informed consent, participants will undergo a cervical stiffness assessment, using the Pregnolia System, followed by their routine pre-induction digital vaginal examination for BS assessment. Participants will then proceed with routine IOL procedure as per the unit policy, either vaginal prostaglandin administration or balloon catheter placement.

Participants will be asked to complete a post-assessment questionnaire at the end of their study visit. This structured questionnaire will collect patient experience of the cervical stiffness assessment in comparison to the BS assessment. This questionnaire data will inform the acceptability of these assessments in this patient population. (appendix 1)

All study participation will coordinate with their planned IOL visit as decided by their clinical team. Participants will remain in the study until after delivery and discharge from hospital. Study outcomes will be collected from electronic hospital records for the participant and their baby. This is outlined in the SPIRIT schedule (Fig 1) and study flow chart (Fig 2).

### Study objectives

The CASPAR study has the following objectives.

1. To inform for the design of an appropriately powered study to assess the capability of this novel device for IOL prediction.

2. To explore the acceptability of the cervical stiffness assessment in patients undergoing IOL.

3. To obtain cervical stiffness measurements in primiparous women prior to term IOL to:

   a. Determine the reliability and best interpretation of triplicate measurements in this patient group.

   b. Explore any potential association between cervical stiffness assessment and vaginal delivery following IOL.

| Procedure | | Induction of labour Booking | Attendance for induction of labour | Postnatal | End of Study |
|---|---|:---:|:---:|:---:|:---:|
| Patient Eligibility | | X | X | | |
| Patient information Leaflet provided | | X | X | | |
| Informed Consent | | X | X | | |
| Routine Assessments | Fetal presentation | | X | | |
| | CTG | | X | | |
| Cervical Stiffness assessment | | | X | | |
| Bishop's score assessment | | | X | | |
| Post- assessment patient questionnaire | | | X | | |
| Primary Outcome | | | | X | |
| Secondary outcomes | | | | X | |
| Adverse Event Reporting | | | X | X | X |

**Fig 1. SPIRIT schedule.**

 c. Compare to BS assessments taken concurrently and explore the association with outcome of IOL.

## Study population

The CASPAR study will consist of consecutive pregnant women attending for IOL at Liverpool Women's Hospital. Study eligibility will be determined on arrival to the IOL suite by the clinical team. Patient's fulfilling the inclusion and exclusion criteria, outlined below, who wish to take part in the study must provide written informed consent for study procedures and use of their data from electronic medical records.

## Inclusion and exclusion criteria

All participants should meet the following inclusion criteria to be eligible:

- Age ≥ 18 years

- Being induced

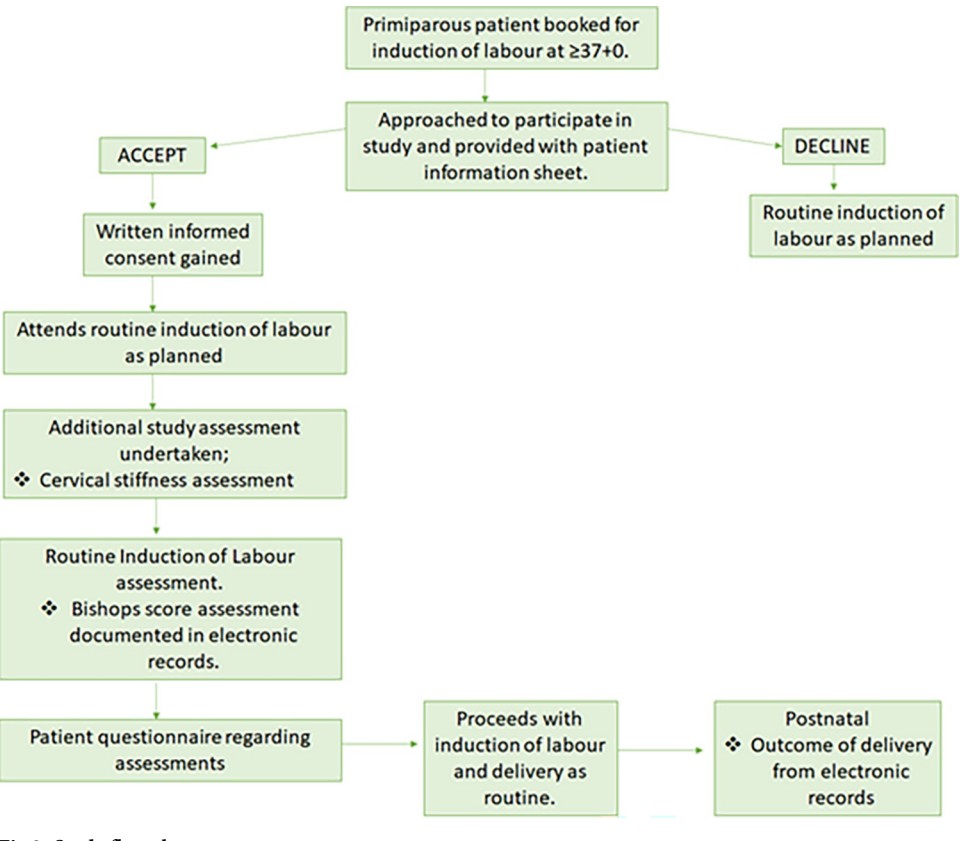

**Fig 2. Study flow chart.**

- Singleton pregnancy

- Primiparous

- ≥37+0 weeks gestation

- Intact membranes

- Able to provide informed consent

    Any possible exclusion criteria will be evaluated for eligibility for the study and includes:

- Previous cervical surgery including previous trachelectomy, cone biopsy, loop excision or previous cerclage

- Any cervical pathology at 12 o'clock position on cervix position (cervical scarring/Nabothian cyst/polyp/cervical tears/cervical myomas/cervical condylomas/cervical endometriosis/cervical cancer)

- Vaginal bleeding evident on examination

- Visible, symptomatic cervical or vaginal infections

- Known congenital uterine anomalies

- Known or suspected structural/chromosomal fetal abnormality

- Known HIV infection

## Study procedures

**Pregnolia system.**  The Pregnolia System is a novel medical device designed to quantitatively assess the biomechanical properties of the cervix through cervical stiffness. The system consists of a single-use sterile probe and a control unit as seen in Fig 3. The control unit is the active component composed of a power supply, foot switch for clinician control, and an integrated pump that generates a vacuum. The sterile probe is attached to the control device via a connector cable and air filters on the probe prevent microbiological contamination [32].

The device requires the use of a vaginal speculum to clearly visualise the cervix. The device probe is placed at the 12 o'clock position on the cervix and the cervical stiffness measurement is operated by a foot switch, creating a weak vacuum which displaces the cervical tissue into the probe tip to a depth of 4mm. The softer the tissue, the less pressure required to deform the tissue [33]. The result is represented by the CSI shown on the control unit. The measurement is audio-guided for the clinician and indicates when the measurement has started, in progress and completed.

The device has instructions for use that outline appropriate clinical indications, contraindications for use and clear instructions. All clinicians using the device will have read the instructions and completed the Pregnolia training instructional video.

**Cervical stiffness assessment.**  Cervical stiffness will be measured using the Pregnolia System. Following informed consent, with the woman in a supine position, the cervix is visualised by use of a sterile speculum and the single-use, sterile Pregnolia Probe is placed on the anterior lip of the cervix at 12 o'clock position. A recording of cervical stiffness is generated over maximum 60 seconds (typically ~15 seconds) and displayed as CSI in mbar. The measurement is repeated 3 consecutive times without any time lag. The clinician is audio-guided by the control unit which indicates when a reading has been taken.

The cervical stiffness readings will be stored on the Pregnolia Control Unit and documented for the participant once all study procedures have been completed. The cervical stiffness readings are blinded to the patient and the clinician who will perform the Bishop's score assessment at the time of the study visit.

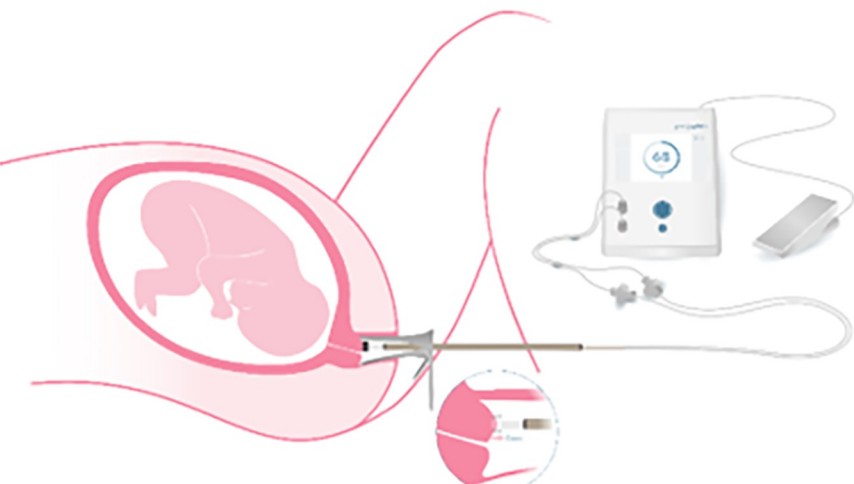

**Fig 3. Pregnolia device- control unit with foot switch and single-use sterile probe.**

Any difficulty in obtaining readings will be documented on the case report form (CRF). Simple troubleshooting guidelines will be followed to optimize readings at the time of the study visit as per Pregnolia instructions (e.g avoiding kinks in connector cable and ensuring Luer locks are secure).

**Bishops score assessment.**   BS assessment will be undertaken by a sterile digital vaginal examination. The study site uses an electronic maternity notes software, and the BS is documented in a standard format. Five components of the vaginal examination will be documented as a score: cervical dilatation (cm), consistency of cervix, cervical canal length (cm), position of cervix and station of presenting part in relation to ischial spines, giving a potential score from 0 to 12.

This procedure will be undertaken by a member of the induction suite midwifery team as per routine unit practice, and their score will be documented in the patient's electronic maternity records. The BS will be taken from the medical records and transferred to the CRF.

## Study outcomes

**Feasibility outcomes.**   Our feasibility outcomes of interest relate to whether the study procedure is acceptable, participant recruitment is achieved, data collection is feasible, and cervical stiffness assessment fidelity is maintained with adequate reliability and safety [34].

Our feasibility outcomes will be defined as follows;

1. Recruitment rate

    a. Measured as proportion of participants recruited compared to total number approached for recruitment in the study period

    b. Reasons for non-participation collected

2. Participant acceptability of cervical stiffness assessment at IOL

    a. Qualitative questionnaire following assessment

    b. Number of participant withdrawals throughout the study duration

3. Adherence to protocol and data collection

    a. Measured as number of protocol deviations

    b. Missing data

4. Cervical stiffness fidelity

    a. Ability to obtain triplicate cervical stiffness measurements

    b. Triplicate measurement reliability

**Clinical outcomes.**   Clinical outcome data will be collected in accordance with the "short-term" core outcome set for trials on IOL as determined by the international Delphi study [35,36]. Ability to capture this core data set in this feasibility study will inform data collection and study design for a larger definitive IOL study.

The primary clinical outcome will be vaginal delivery. Secondary maternal and neonatal outcomes will be recorded for descriptive analyses. Variables that are already recognised as informative for induction prediction modelling will be collected and included for descriptive analyses such as; maternal demographics, maternal obstetric parameters, induction methods and indications, cervical findings on digital examination and fetal parameters [8,23].

## Study endpoint

The study will end when the last recruited woman has delivered and both herself and her baby have been discharged from hospital, or 1 month after delivery, whichever is sooner and all planned analysis of collected data has taken place.

## Statistical analysis

**Sample size.** The Pregnolia System is a novel device with no published data providing cervical stiffness index results in primiparous women undergoing IOL. This will be the first study to provide this data and will guide how further research using this novel assessment tool can be most appropriately designed.

In this feasibility study we will recruit for 12 months and expect to achieve a cohort of 100 participants. The study site has a high IOL capacity with on average 100 primiparous women being induced every month. We anticipate 50% to be eligible for the study and aim to achieve 50% recruitment. Previous IOL trials have struggled with recruitment, with over 70% declining participation [37]. However, in our feasibility study design the clinical indication and timing of IOL have already been determined and the participant is already expecting to undergo an IOL at the time of study recruitment. The study does not influence or change routine clinical practice therefore, we expect to achieve a moderate recruitment rate of 50%. Clearly determining the expected recruitment rate for this novel device is of paramount importance for future successful definitive study design.

At the study site, routine IOL appointments are from 8am-8pm with emergency indications being accommodated throughout the 24hr, 7-day service. In this feasibility study we understand that not all eligible women will be able to be approached given the 24hr 7-day nature of the service. We have planned for a 12-month recruitment period to allow for these challenges and aim to achieve a sample size of 100 participants. Ability to achieve this target number for this feasibility study with the small study team available will help inform the larger study design resource and team allocation for study recruitment and study visit procedures in this dynamic clinical setting.

This feasibility study will provide key data including an accurate recruitment rate demonstrating this population's acceptability for the novel assessment in clinical practice, and an estimate of the variance for cervical stiffness using the Pregnolia System for pre-induction cervical assessment. Together this data can infer a larger study design for a predictive model for IOL outcome in this population [38–40].

**Statistical analysis.** Descriptive statistics will be generated and presented as means (SD), median (IQR) and frequency of observations (percentages) with 95% confidence intervals as appropriate.

Reliability assessment of the Pregnolia System using Cronbach's alpha and descriptive statistics with 95% confidence intervals will inform best use of the triplicate CSI measurements. Specifically exploring whether the first, average, median or lowest measurement of the three readings should be utilised in further analysis and most importantly inform best use of the CSI results in clinical practice.

Diagnostic performance of cervical stiffness assessments using the Pregnolia System for IOL outcome will be demonstrated through receiver operating characteristic (ROC) curves with area under the curve and 95% confidence intervals being calculated, as well as aiming to define the optimum cut-off value for predicting vaginal delivery. Alongside the primary clinical outcome of mode of delivery, other IOL outcomes of interest will include oxytocin requirement and duration, requirement for artificial rupture of membranes and total duration of rupture membranes, caesarean section for failed IOL, achieving established labour and >1 cervical ripening method required. Bishop's score diagnostic performance data will also be generated for direct comparison.

If a positive signal between CSI and an IOL outcome is achieved, a multivariate analysis will be performed using logistic regression, including CSI and other variables related to IOL outcome, such as maternal age, gestational age and maternal weight at booking [23]. Results will be presented as odds ratios with 95% confidence intervals.

Acceptability of the device will be determined using participant questionnaires. Results will be collated and analysed providing descriptive statistics presented as means (SD), median (IQR) and frequency of responses (percentages). The participant experience of each study procedure is collected as a scale response from 1–10. A paired t-test with 95% confidence intervals will be used for comparison of participant experience between the different study procedures. A p value of <0.05 will determine significance.

## Data management plan

All study participants are allocated a unique participant identification number and all data relating to that patient is pseudonymised. Study data is initially captured using paper CRF. The data from the paper CRFs is then transcribed to an electronic CRF within a bespoke, password protected, study database (REDCAP). The Chief Investigator (CI) preserves the confidentiality of participants taking part in the study and abides by the EU General Data Protection Regulation 2016 and Data Protection Act 2018.

**Consent and criteria for withdrawal.** Consent to enter the study must be sought from each participant only after a full explanation has been given, an information leaflet offered, and time allowed for consideration. Signed participant consent will be obtained. The participant can decline to participate without giving reasons and this will not impact upon further care. All participants are free to withdraw at any time from the study without giving reasons and without prejudicing further care.

In addition, the CI may decide, for reasons of medical prudence, to withdraw a participant. In either event, the Sponsor will be notified and the date and reason(s) for the withdrawal will be documented in the participant source data. If a participant withdraws or is withdrawn, ideally, they should remain in the study for the collection of outcome data. If the participant states their wish not to contribute further data to the study, collected data will be removed from the study database and no further outcome date will be collected.

**Monitoring and safety.** A Study Management Group (SMG) comprising the CI, principal investigator, co-applicants, and core study management staff meet at regular intervals throughout the course of the study and holds responsibility for the day-to-day running and management of the study. The need to stop the study will be determined by the SMG and the decision will be based upon data integrity and participant safety.

Any adverse events for this study will be recorded at each study visit on the study CRF. This is a non-interventional study we therefore do not anticipate many serious adverse events. The Pregnolia System is a licensed, CE-marked, non-invasive medical device and the risk of adverse events related to the measurement is small. The device has a very good safety profile, and the manufacturer has not received any serious adverse event reporting related to the medical device either in studies or clinical practice to date.

**Ethical considerations and declarations.** The Seasonal Research Ethics Committee has given approval for this research (23/LO/0627). All patients have given written informed consent prior to entry to the study and are aware that participation is completely voluntary.

**Status and timeline of the study.** This study is registered at ClinicalTrials.gov, identifier NCT05981469 and has been open for recruitment since 29[th] September 2023. Recruitment is due to finish 30[th] September 2024, and patient follow up complete in November 2024.

## Discussion

The CASPAR feasibility study will provide original data using the Pregnolia System for cervical stiffness assessments prior to IOL. Determining the acceptability and efficacy of this novel assessment tool for pre-induction cervical assessment is critical prior to funding and undertaking a larger, powered study incorporating the Pregnolia System assessment tool into a clinical prediction model for IOL.

## Supporting information

**S1 Checklist. SPIRIT 2013 checklist: Recommended items to address in a clinical trial protocol and related documents\*.**
(DOC)

**S1 Fig. Participant questionnaire.**
(PDF)

**S1 File.**
(PDF)

**S2 File.**
(PDF)

**S3 File.**
(DOCX)

## Author Contributions

**Conceptualization:** Andrew Sharp, Angharad Care.

**Data curation:** Elizabeth Medford.

**Formal analysis:** Elizabeth Medford, Steven Lane.

**Funding acquisition:** Angharad Care.

**Methodology:** Elizabeth Medford, Steven Lane, Abi Merriel, Andrew Sharp, Angharad Care.

**Project administration:** Elizabeth Medford.

**Resources:** Angharad Care.

**Supervision:** Andrew Sharp, Angharad Care.

**Writing – original draft:** Elizabeth Medford.

**Writing – review & editing:** Elizabeth Medford, Abi Merriel, Andrew Sharp, Angharad Care.

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
