## [Decision Letter · Decision Letter 0]

19 Nov 2024

PONE-D-24-39289Study Protocol. The CASPAR study protocol. Can cervical stiffness predict successful vaginal delivery after induction of labour? A feasibility, cohort study.PLOS ONE

Dear Dr. Medford,

Thank you for submitting your manuscript to PLOS ONE. After careful consideration, we feel that it has merit but does not fully meet PLOS ONE’s publication criteria as it currently stands. Therefore, we invite you to submit a revised version of the manuscript that addresses the points raised during the review process.

ACADEMIC EDITOR: Please respond to all reviewers comments==============================

We look forward to receiving your revised manuscript.

Kind regards,

Ahmed Mohamed Maged, MD

Academic Editor

PLOS ONE

Journal Requirements: When submitting your revision, we need you to address these additional requirements. 1. Please ensure that your manuscript meets PLOS ONE's style requirements, including those for file naming. The PLOS ONE style templates can be found at https://journals.plos.org/plosone/s/file?id=wjVg/PLOSOne_formatting_sample_main_body.pdf and https://journals.plos.org/plosone/s/file?id=ba62/PLOSOne_formatting_sample_title_authors_affiliations.pdf 2. We note that the original protocol that you have uploaded as a Supporting Information file contains an institutional logo. As this logo is likely copyrighted, we ask that you please remove it from this file and upload an updated version upon resubmission. 3. Please ensure that you refer to Figure 3 in your text as, if accepted, production will need this reference to link the reader to the figure.

Reviewers' comments:

Reviewer's Responses to Questions

**Comments to the Author**

1. Does the manuscript provide a valid rationale for the proposed study, with clearly identified and justified research questions?

Reviewer #1: Yes

Reviewer #2: Yes

2. Is the protocol technically sound and planned in a manner that will lead to a meaningful outcome and allow testing the stated hypotheses?

Reviewer #1: Yes

Reviewer #2: Yes

3. Is the methodology feasible and described in sufficient detail to allow the work to be replicable?

Reviewer #1: Yes

Reviewer #2: Yes

4. Have the authors described where all data underlying the findings will be made available when the study is complete?

Reviewer #1: Yes

Reviewer #2: Yes

5. Is the manuscript presented in an intelligible fashion and written in standard English?

Reviewer #1: Yes

Reviewer #2: Yes

6. Review Comments to the Author

You may also provide optional suggestions and comments to authors that they might find helpful in planning their study.

Reviewer #1: According to the investigators , a feasibility study is needed to explore the application of this novel device to the IOL setting for pre induction cervical assessment. Also, statistically assessing the acceptability, implementation and efficacy of the Pregnolia System for consistent cervical stiffness assessments in this setting may be practical. CASPAR is a feasibility study of term, primiparous women with singleton pregnancies undergoing IOL. A protocol has been formulated with this goal.

The study is well designed from sample size and implementation perspective and the statistical analysis plan is presented with most of the essential detail. Section 6.0 in the supporting information Appendix provides such detail. Outcome measures are seen in another section 3.2. There are two minor clarifications needed in section 6.5 , statistical methodology. Specific IOL responses as well as variable inputs should be added to the multivariate logistic model or models in this paragraph. Also several of the questions asked of the subject (especially for satisfaction) have a likert scale format. Some specific non parametric analyses should be mentioned in this section as well.

Reviewer #2: Well designed study protocol; Included all the practical aspects for the feasibility study; Patient questionnaire covering all the aspects for the assessment from their perspective

7. PLOS authors have the option to publish the peer review history of their article (what does this mean?). If published, this will include your full peer review and any attached files.

Reviewer #1: No

Reviewer #2: **Yes: **sasirekha rengaraj

---

## [Author Response · Author response to Decision Letter 0]

13 Dec 2024

Dear Editors and Academic Reviewers, 

Thank you for taking the time to review my manuscript; “The CASPAR study protocol. Can cervical stiffness predict successful vaginal delivery after induction of labour? A feasibility, cohort study.” 

Please find below our response to each individual point raised. 

 Author response: I believe my manuscript meets the PLOS ONE style requirements. 

2. We note that the original protocol that you have uploaded as a Supporting Information file contains an institutional logo. As this logo is likely copyrighted, we ask that you please remove it from this file and upload an updated version upon resubmission.

 Author response: The logo included in the original protocol is a CASPAR specific logo and designed by the team that has designed this study. This image is not copyrighted. However, for ease I have removed the image from this document as requested and uploaded the updated protocol as supporting information. 

3. Please ensure that you refer to Figure 3 in your text as, if accepted, production will need this reference to link the reader to the figure.

Author response: Thank you for highlighting this error. Amendment made line 165, the manuscript now correctly references figure 3. 

Reviewer’s comments:

Reviewer #1: According to the investigators , a feasibility study is needed to explore the application of this novel device to the IOL setting for pre induction cervical assessment. Also, statistically assessing the acceptability, implementation and efficacy of the Pregnolia System for consistent cervical stiffness assessments in this setting may be practical. CASPAR is a feasibility study of term, primiparous women with singleton pregnancies undergoing IOL. A protocol has been formulated with this goal.

The study is well designed from sample size and implementation perspective and the statistical analysis plan is presented with most of the essential detail. Section 6.0 in the supporting information Appendix provides such detail. Outcome measures are seen in another section 3.2. There are two minor clarifications needed in section 6.5 , statistical methodology. Specific IOL responses as well as variable inputs should be added to the multivariate logistic model or models in this paragraph. Also several of the questions asked of the subject (especially for satisfaction) have a likert scale format. Some specific non parametric analyses should be mentioned in this section as well.

Author response: Thank you for your review. We aim to be as transparent as possible regarding our outcomes and statistical analysis plan. 

Section 3.2 in the protocol (supporting information) is included in the manuscript lines 204- 237. 

Section 6.5 in the protocol (supporting information) has been included in the manuscript from lines 272-284. 

Amended line 281-285 to included further specific IOL responses of interest. Amended line 286 to reiterate that a multivariate analysis will only be undertaken if a positive signal for cervical stiffness and IOL outcome is identified. 

Thank you for highlighting that the statistical plan for questionnaire analysis had not been included. Amendments made, line 290-295. 

Reviewer #2: Well designed study protocol; Included all the practical aspects for the feasibility study; Patient questionnaire covering all the aspects for the assessment from their perspective

Author response: Thank you for your review. 

Thank you for your consideration of my revised manuscript. I look forward to hearing the outcome. If you require any further information or clarification, please do not hesitate to contact me. 

Kind regards, 

Elizabeth Medford

---

## [Decision Letter · Decision Letter 1]

27 Dec 2024

Study Protocol. The CASPAR study protocol. Can cervical stiffness predict successful vaginal delivery after induction of labour? A feasibility, cohort study.

PONE-D-24-39289R1

Dear Dr. Medford,

We’re pleased to inform you that your manuscript has been judged scientifically suitable for publication and will be formally accepted for publication once it meets all outstanding technical requirements.

Kind regards,

Ahmed Mohamed Maged, MD

Academic Editor

PLOS ONE

Additional Editor Comments (optional):

Reviewers' comments:

Reviewer's Responses to Questions

**Comments to the Author**

1. Does the manuscript provide a valid rationale for the proposed study, with clearly identified and justified research questions?

Reviewer #1: Yes

2. Is the protocol technically sound and planned in a manner that will lead to a meaningful outcome and allow testing the stated hypotheses?

Reviewer #1: Yes

3. Is the methodology feasible and described in sufficient detail to allow the work to be replicable?

Reviewer #1: Yes

4. Have the authors described where all data underlying the findings will be made available when the study is complete?

Reviewer #1: Yes

5. Is the manuscript presented in an intelligible fashion and written in standard English?

Reviewer #1: Yes

6. Review Comments to the Author

You may also provide optional suggestions and comments to authors that they might find helpful in planning their study.

Reviewer #1: All comments have been addressed and changes incorporated into the document.

The model has been specified and variables included as suggested.

7. PLOS authors have the option to publish the peer review history of their article (what does this mean?). If published, this will include your full peer review and any attached files.

Reviewer #1: No

---

## [Editor Report · Acceptance letter]

7 Jan 2025

PONE-D-24-39289R1 

PLOS ONE

Dear Dr. Medford, 

I'm pleased to inform you that your manuscript has been deemed suitable for publication in PLOS ONE. Congratulations! Your manuscript is now being handed over to our production team.

Kind regards, 

on behalf of

Professor Ahmed Mohamed Maged 

Academic Editor

PLOS ONE